# Molecular basis for the phosphorylation of bacterial tyrosine kinase Wzc

Yun Yang[1,2,9], Mariana Batista[3,9], Bradley R. Clarke [4], Michelle R. Agyare-Tabbi[4], Haigang Song [5,6], Noah M. Kuehfuss[4], Audrey Le Bas[1], Carol V. Robinson [5,6], Chris Whitfield [4] ✉, Phillip J. Stansfeld [3] ✉, James H. Naismith [1,7,8] ✉ & Jiwei Liu [1,2] ✉

The regulation of polymerisation and translocation of biomolecules is fundamental. Wzc, an integral cytoplasmic membrane tyrosine autokinase protein serves as the master regulator of the biosynthesis and export of many bacterial capsular polysaccharides and exopolysaccharides. Such polysaccharides play essential roles in infection, defence, and some are important industrial products. Wzc comprises a large periplasmic domain, two transmembrane helices and a C-terminal cytoplasmic kinase domain with a tyrosine-rich tail. Wzc regulates polymerisation functions through cycling the formation and dissociation of an octameric complex, driven by changes in the phosphorylation status of the tyrosine-rich tail. *E. coli* Wzc serves a model for a wider family of polysaccharide co-polymerases. Here, we determine structures of intermediate states with different extents of phosphorylation. Structural and computational data reveal the pre-ordering of the tyrosine-rich tail, the molecular basis underlying the unidirectionality of phosphorylation events, and the underlying structural dynamics on how phosphorylation status is transmitted.

Many bacterial capsular and secreted extracellular polysaccharides (CPS and EPS) are assembled by a "Wzy-dependent" strategy (reviewed in ref. 1; Fig. 1a, b). These polymers are vital to bacteria as they are involved in immune evasion, stress response and antibiotic resistance. Consequently, the synthesis and export of CPS and EPS represents a potential target for antibacterial intervention[1,2]. In Wzy-dependent pathways, Wzx is a flippase belonging to the multi-antimicrobial extrusion protein family, which translocates undecaprenol diphosphate-linked polysaccharide repeat units produced at the cytoplasm-membrane interface across the cytoplasmic membrane[3]. Once available at the periplasmic face of the membrane, the lipid intermediates provide substrates for the Wzy polymerase, a glycosyltransferase possessing a GT-C fold[4] which extends the nascent polysaccharide chain by adding one repeat unit at a time. The polymerisation reaction performed by Wzy is regulated by a polysaccharide co-polymerase protein (PCP). There are two classes of PCP, exemplified by Wzz (PCP-1, regulating lipopolysaccharide O-antigen polysaccharide polymerization) and Wzc (PCP-2, regulating polymerization of CPS and EPS)[1,5]. The Wzy and Wzz proteins from *Pectobacterium atrosepticum* have been shown to form a complex of one Wzy with eight Wzz proteins[6] and it is expected that Wzy and their cognate PCP form a complex.

[1]Structural Biology, The Rosalind Franklin Institute, Harwell Campus, Didcot, UK. [2]West China School of Public Health and West China Fourth Hospital, and State Key Laboratory of Biotherapy, Sichuan University, Chengdu, China. [3]School of Life Sciences & Department of Chemistry, University of Warwick, Coventry, UK. [4]Department of Molecular and Cellular Biology, University of Guelph, Guelph, Ontario, Canada. [5]Kavli Institute for NanoScience Discovery, Dorothy Crowfoot Hodgkin Building, University of Oxford, Oxford, UK. [6]Department of Chemistry, University of Oxford, Oxford, UK. [7]The Mathematical, Physical and Life Sciences Division, University of Oxford, Oxford, UK. [8]Division of Structural Biology, Roosevelt Drive, The University of Oxford, Oxford, UK. [9]These authors contributed equally: Yun Yang, Mariana Batista. ✉e-mail: cwhitfie@uoguelph.ca; phillip.stansfeld@warwick.ac.uk; naismith@strubi.ox.ac.uk; jiwei.liu@rfi.ac.uk

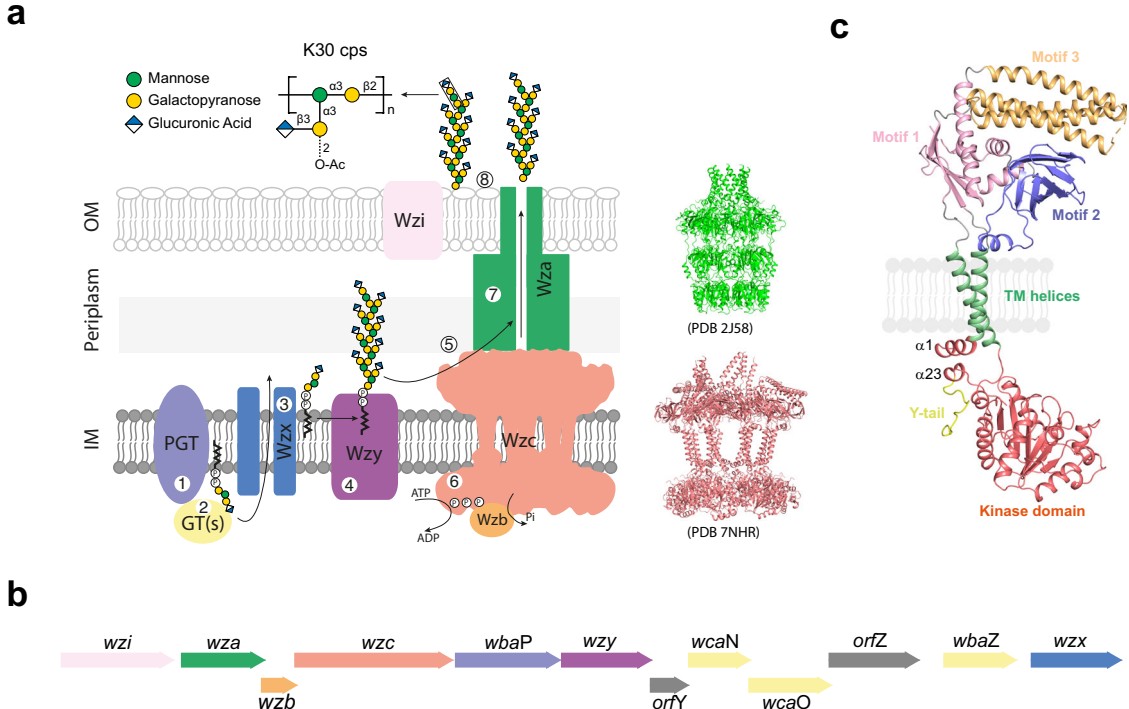

**Fig. 1 | Wzc and the Wzx-Wzy pathway. a** Schematic model for the assembly and export of capsular polysaccharide (CPS) in *E. coli* K30, the prototype for Wzy-dependent CPS biogenesis. The repeat unit of *E.coli* K30 CPS[50] is synthesized on an undecaprenol diphosphate carrier lipid at the cytoplasmic side of the inner membrane (IM) by a phosphoglycosyltransferase (PGT; WbaP) and glycosyltransferases (GT(s); WbaZ, WcaO and WcaN), and flipped to the periplasmic face of the membrane by Wzx. The repeat units are polymerized by Wzy, and the polymers are exported by Wza[1]. Polymerization and export are regulated by Wzc (a PCP-2a family autokinase), which cycles between a phosphorylated state and dephosphorylated state due the activity of the Wzb phosphatase[9]. Wzi is involved in the surface attachment of CPS[1]. **b** The gene cluster for the synthesis and export of the CPS of *E. coli* K30 (GenBank Acc. AF104912.3). **c** A cartoon of a Wzc monomer, comprising the periplasmic domain containing motif 1–3, two transmembrane helices, the BY-kinase domain, and the C-terminal tyrosine-rich tail (Y-tail) (adapted from[9]).

Wzc PCP-2 proteins found in CPS and EPS biosynthetic pathways are integral membrane proteins with two transmembrane helices (TM) that flank a large periplasmic domain[7–10] (Fig. 1c). The second of the two helices connects to a cytoplasmic bacterial tyrosine (BY-) kinase catalytic domain (containing characteristic Walker A and B nucleotide binding motifs), followed by a C-terminal tyrosine-rich tail[7–10]. The kinase domain distinguishes PCP-2 proteins from PCP-1 proteins; the kinase mechanism is the focus of this study. Tyrosine kinases were originally thought to be confined to eukaryotes, but are now known to be prevalent among bacteria where they play a variety of roles[11]. BY-kinases are distinct in both structure and sequence from eukaryotic kinases[8], enhancing their potential as antibacterial targets. In Gram-positive bacteria, the transmembrane and cytoplasmic BY-kinase domains of PCP-2 proteins exist as two polypeptides[1,12,13] that form a complex which appears to function in the same manner as the single chain Wzc found in Gram-negative bacteria. The BY-kinase domain is highly conserved and Wzc proteins (and their Gram-positive paralogues) almost all possess a C-terminal tyrosine-rich tail[7]. Both the phosphorylation and dephosphorylation of the Wzc C-terminal tyrosine-rich tail were shown to be required for polymerisation and export, presumably by regulation of the polymerase Wzy and possibly other proteins[9,14]. Dephosphorylation of Wzc is catalysed by a cognate phosphatase (Wzb a protein tyrosine phosphatase (PTP) family member) in Gram-negative bacteria, and by a protein histidinol phosphatase (PHP) in Gram-positive bacteria[15]. Mutations eliminating phosphatase function result in drastically reduced CPS production, phenocopying kinase-null mutations[14]. Mutagenesis of the tyrosine-rich tail established that the regulatory function of Wzc in *E. coli* CPS biosynthesis is determined by the extent of phosphorylation rather than any individual tyrosine residue[9,14,16,17].

PCP proteins form oligomers and we previously reported the cryo-EM structures of octamers of the dephosphorylated form of Wzc from *E. coli* serotype K30 (the CPS assembly prototype) by using an inactive (kinase) mutant, K540M[9]. A large central cavity in the octamer is formed by the transmembrane helices (Fig. 1)[9]. The periplasmic domain has three motifs (numbered 1, 2 and 3), with motif 3 proposed to interact with the outer membrane CPS translocon formed by an octamer of protein Wza[9,18]. Motif 3 varies in its structural arrangement between monomers (breaking the eight-fold symmetry), and in some monomers motif 3 is significantly disordered[9]. The C-terminal tail has a tyrosine-rich portion (Y705 to K721) which inserts into the active site of the neighbouring protomer[9]. A similar insertion of the tyrosine tail that stabilises an octameric ring also seen in the isolated BY-kinases from *Staphylococcus aureus*[13] and *E.coli* K-12[19], indicating a conserved feature. Mass spectrometry data demonstrated that tyrosine phosphorylation of Wzc from *E. coli* started at either Y718 (not conserved) or Y717 (highly conserved) and progressed sequentially to further tyrosine residues towards the N-terminus. Sequential phosphorylation destabilised the oligomers and Wzc[K540M]4YE (K540M, Y718E, Y717E, Y715E and Y713E) was engineered to resemble the phosphorylated state at the observed stability "tipping-point". Introduction of an additional mutation (Y708E) resulted in disassembly of the octamer[9]. An NMR study[20] implicated residues 508–514 in recognition by the cognate phosphatase, Wzb[15]. Since the cryoEM structures indicated these residues are not fully accessible even in Wzc[K540M]4YE mutant (i.e. the least stable octamer)[9], we hypothesize that the phosphatase may only be able to access its substrate once the octamer has disassembled. We proposed a model in which the assembly and disassembly of the octamer around the Wzy polymerase was at the heart of the molecular basis of the regulation of Wzc[9].

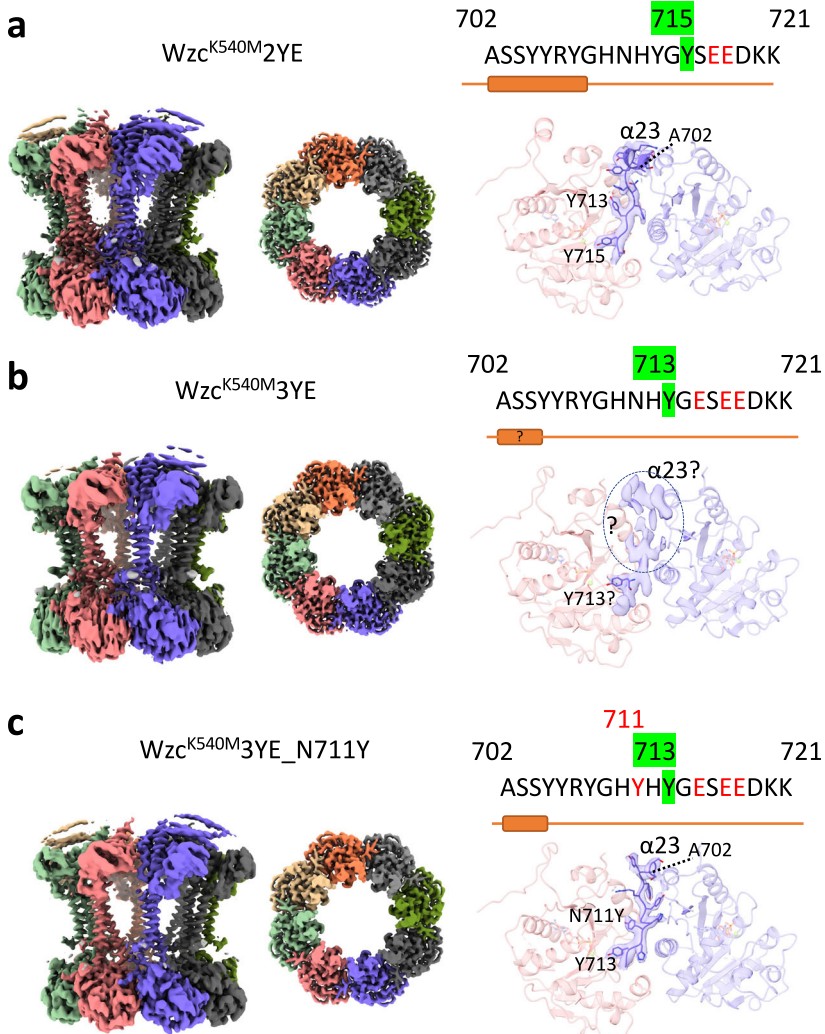

**Fig. 2 | Cryo-EM maps of Wzc mutants. a** Wzc^K540M^2YE. **b** Wzc^K540M^3YE. **c** Wzc^K540M^3YE_N711Y. In each panel, on the left is shown the coulombic map of the entire molecule viewed from the side. The middle image shows the coulombic map of kinase domains only, viewed from the periplasm. The coulombic map of helix α23 and the tyrosine tail as they insert into the active site of the neighbouring kinase is shown on the right.

While the reported structures offered important insight into the function of Wzc, there remained significant gaps in our molecular understanding, including structural changes that occur during phosphorylation, the determinants of phosphorylation directionality, and regulatory activity.

Here, we report the characterisation of the structures Wzc^K540M^2YE (Y718E, Y717E), Wzc^K540M^3YE (Y718E, Y717E and Y715E) and Wzc^K540M^3YE_N711Y (an additional N711Y mutation) corresponding to different phosphorylation states. These structures are used to develop a molecular dynamics model that rationalises the directionality of the phosphorylation, the dynamics of the C-terminal tail, and transmission of information on the phosphorylation state.

## Results

### Experimental structural snapshots

In Wzc from *E. coli* K30, tyrosine residues 708, 713, 715, 717 and 718 are targets for phosphorylation; residue Y718 is neither conserved in Gram-negative PCP-2 proteins, nor is it essential for function in CPS assembly or downstream phosphorylation[9]. We used mutation of tyrosine to glutamate to mimic phosphorylation and the inactive K540M mutant was used for structural studies as previously described[9]. The Wzc^K540M^2YE (Y718E and Y717E) mutant corresponds to a doubly phosphorylated form but includes the first conserved and

functional phosphorylation event (denoted 1pY). At a global level, the structure (periplasmic, transmembrane and cytoplasmic domains) (Fig. 2a, Supplementary Figs. 1, 4) is very similar to the octameric Wzc^K540M^ structure. Briefly, portions of the periplasmic domains do not follow the 8-fold symmetry that is seen in the membrane spanning and cytoplasmic domains. In three subunits, four helices are folded into a bundle that sits on motif 1 from the periplasmic domain. In four subunits, these four helices are mostly disordered, but the ordered portions are oriented parallel to the membrane normal. In one subunit, these helices are entirely disordered (Fig. 2a, Supplementary Figs. 1, 4). Each of these configurations and their arrangement in the octamer are identical to those seen in Wzc^K540M^ structure.

The remainder of the periplasmic domain is arranged around the central 8-fold axis, creating a large central cavity (Fig. 2a, Supplementary Figs. 1, 4). The transmembrane region has eight pairs of helices arranged as a circle enclosing a central void. The transmembrane helices are not close packed, leaving large portals open to the membrane bilayer. The cytoplasmic BY-kinase domains form a circular arrangement with the C-terminal tyrosine-rich tail reaching into the neighbouring protomer. The presence of ADP and MgCl$_2$ improved the quality of the coulombic map for Wzc^K540M^2YE, permitting unambiguous identification of Y715 at the (neighbouring) kinase active site

(Fig. 2a, Supplementary Fig. 1). As a consequence, the interaction between the two protomers is different than previously observed[9]. In order for Y715 to bind at the catalytic site (as opposed to Y717 seen in Wzc[K540M] structure[9]) the loop provided by residues 710–714 has changed conformation to become a strand. Y713 sits in the same pocket that contains Y715 in the published Wzc[K540M]structure[9]. This "tyrosine pocket" is formed by loops A534-P536, P645-L647 and residues E675 and S679 (Supplementary Fig. 5d). The arrangement of the residues that form the pocket are identical in the two structures (one with Y715 and the other with Y713 in the pocket) (Supplementary Fig. 5d). The positions of the catalytic site residues and the kinase domain residues that interact with the tail do not however show (at the resolution of our studies) any change in position or conformation.

The Wzc[K540M]3YE (Y718E, Y717E and Y715E) mutant was designed to mimic the conserved doubly phosphorylated species (2pY) and we therefore expected Y713 at the catalytic site. Although a tyrosine was clearly at the kinase active site, poor quality coulombic density unfortunately precluded its unambiguous identification (Fig. 2b, Supplementary Fig. 2). The disorder seen in the C-terminal tail, beginning at S704, was not observed in any other structure (Wzc[K540M], Wzc[K540M]4YE[9], Wzc[K540M]2YE (this study)). No residue was visible in the tyrosine pocket, if Y713 is at the catalytic site then N711 would be expected to be at the pocket. We hypothesised that the pocket does not bind an asparagine side chain, and that leads to the observed disorder. To test this, we further mutated Wzc[K540M]3YE to introduce N711Y (Wzc[K540M]3YE_N711Y). The resulting structure confirmed our hypothesis; it possessed an ordered tail that unambiguously identified Y713 at the catalytic site with N711Y in the tyrosine pocket (Fig. 2c, Supplementary Fig. 3). The relative positions of N711Y and the pocket are different than that observed in Wzc[K540M] and Wzc[K540M]2YE but are almost identical to that previously described in Wzc[K540M] 4YE (Fig. 2c, Supplementary Fig. 5d). The resolution of the EM structures and the flexibility of this region precludes definitive analysis of specific molecular interactions. As a result, it was not possible to convincingly rationalise the preference of the pocket for tyrosine over asparagine, except to note the aromatic ring of tyrosine would make interactions with the hydrophobic ring of Pro 536 that are not possible for asparagine. To accommodate Y713 at the catalytic site and N711Y in the pocket, helix α23 has partially unwound to a strand at its C-terminus (Y705). In the Wzc[K540M]3YE structure, this region of the helix is disordered. Excluding the disorder/order of the C-terminus, the structure of Wzc[K540M]3YE_N711Y was essentially unchanged relative to Wzc[K540M]3YE (Supplementary Fig. 6). The four helices in the periplasmic domain show the same pattern of structures that was observed in Wzc[K540M] and Wzc[K540M]2YE. More broadly, the overall structure of the catalytic site and that of the kinase domain are identical in all the experimental structures.

With these additional structures, a complete set of snapshots now exists for different (phosphorylation) states of the octamer: Wzc[K540M] (0pY), Wzc[K540M]2YE (1pY), Wzc[K540M]3YE (2pY), and Wzc[K540M]4YE (3pY) (Fig. 3a, Supplementary Fig. 4), Wzc[K540M]5YE (4pY) does not form a stable octamer[9]. Comparison of the structures reveals that the differences within the tail and kinase active site are accompanied by a progressive rotation of the transmembrane and periplasmic domains by 3° (relative to 0pY) for 1pY, 5° for 2pY and 12° for 3pY (Fig. 3b). The interaction between N-terminal helix α1 and helix α23 (which directly connects to the moving C-terminal tyrosine tail) changes as Wzc progresses through the states (Fig. 3b, c). Relative to the remainder of the structure, the position and structure of these two helices undergo only very small changes between the 0pY and 1pY states (Fig. 3b, c). In 2pY, helix α23 has partially unwound, retaining only one helical turn, and consequently has a diminished interface with α1 (Figs. 2b, c, 3b, c). In 3pY, α23 is completely unwound and retains no interaction with α1[9]. Throughout these changes, the helical conformation of α1 is preserved, with only minor positional shifts observed (Fig. 3b, c). A

rotation of the transmembrane and periplasmic regions, relative to the kinase ring, is observed in comparison of the octamer structures (Fig. 4).

## Molecular dynamics analysis of phosphorylation

To gain insight into transitions between phosphorylation states, we performed a series of MD simulations. Studying the entire molecule was too computationally expensive to be feasible, so we used the soluble cytoplasmic catalytic domains and the tyrosine-rich tails of two adjacent Wzc subunits (Fig. 5a). We modelled four phosphorylation states (0pY, 1pY, 2pY and 3pY) (Supplementary Table 2) and compared these simulations with the experimental structures from the Tyr-to-Glu mutants. The distance between the Cβ of the tyrosine and the Cα of D564 from the adjacent subunit was used as metric for structural changes (Fig. 5a). The simulation reveals that stability of the dimer is inversely correlated with the number of phosphorylated tyrosines. In essence, phosphotyrosines positioned C-terminal to the residue at the active site are progressively destabilising. Repeating the MD calculations with ADP at the active site, we observed the complete displacement of the phosphotyrosine from the active site in each model (Fig. 5b, Supplementary Fig. 7). When Mg[2+] was introduced into the model, it stabilised the position of ADP by coordinating to both ADP and phosphotyrosine. We investigated the Tyr-to-Glu mutants in MD and observed that the mutated residue quickly leaves the active site in all cases (Fig. 5b, Supplementary Fig. 8). This validates our assumption that the substitution is a good mimic for the natural phosphorylation of the tyrosine. After leaving the active site, the negatively charged residue (phosphotyrosine or glutamate) was not observed to bind elsewhere on the structure. The model showed that in order for Y708 to enter the active site, the C-terminal tail pulls helix α23 away from helix α1, resembling closely the structure of 3pY (Wzc[K540M]4YE[9]) (Fig. 5c).

## Mechanism for transmembrane signal transduction

To assess the functional role of the helix α1 and α23 interactions, we performed MD simulation of two neighbouring protomers embedded in a phospholipid bilayer. For these simulations, we selected 0pY (where the interaction is strongest) and 3pY (where the α23 and α1 interactions are absent). To measure the conformational changes, we calculated the angles between each of the helices TMH1 (transmembrane helix 1), TMH2, α1 and the axis perpendicular to the membrane plane, as well as the relative angle between TMH1 and TMH2 (Fig. 6a, b). In the absence of phosphorylation with Y717 at the active site (Fig. 5b), the C-terminal tail around the active site exhibits low flexibility (Fig. 6c, d). In the absence of ADP, pY708 remained at the 3pY active site, resulting in limited flexibility and preservation of a D28-K701 contact that links helices α1 and α23 (Fig. 6b–e). In the presence of ADP pY708 was displaced, leading to increased C-terminal flexibility and disruption of the D28-K701 contact which, in turn, mobilised the α1 helix (Fig. 6b–e). Similar results were observed when simulating three neighbouring protomers in a phospholipid bilayer (Supplementary Figs. 9, 10). Superposition of the initial and final frames of the simulation revealed a structural rearrangement in the protein driven by rotation of the helices α1 and TMH1, which is transmitted to the periplasmic domain (similar observation were made with Wzc[K540M]4YE) (Fig. 6d). MD analysis shows that phosphorylation of the tail induces conformational changes in transmembrane and periplasmic domains via direct coupling between helices α23 and α1. To probe the D28-K701 contact, mutants were created (D28A, K710A and D28A/K701A) and all mutants resulted in a drastic reduction in CPS production (Fig. 6f) confirming the importance of this contact.

## Discussion

The assembly and disassembly of the Wzc octamer is driven by phosphorylation and dephosphorylation of the C-terminal tyrosine-rich tail[9,14] and this cycling underpins the function of Wzc in regulating

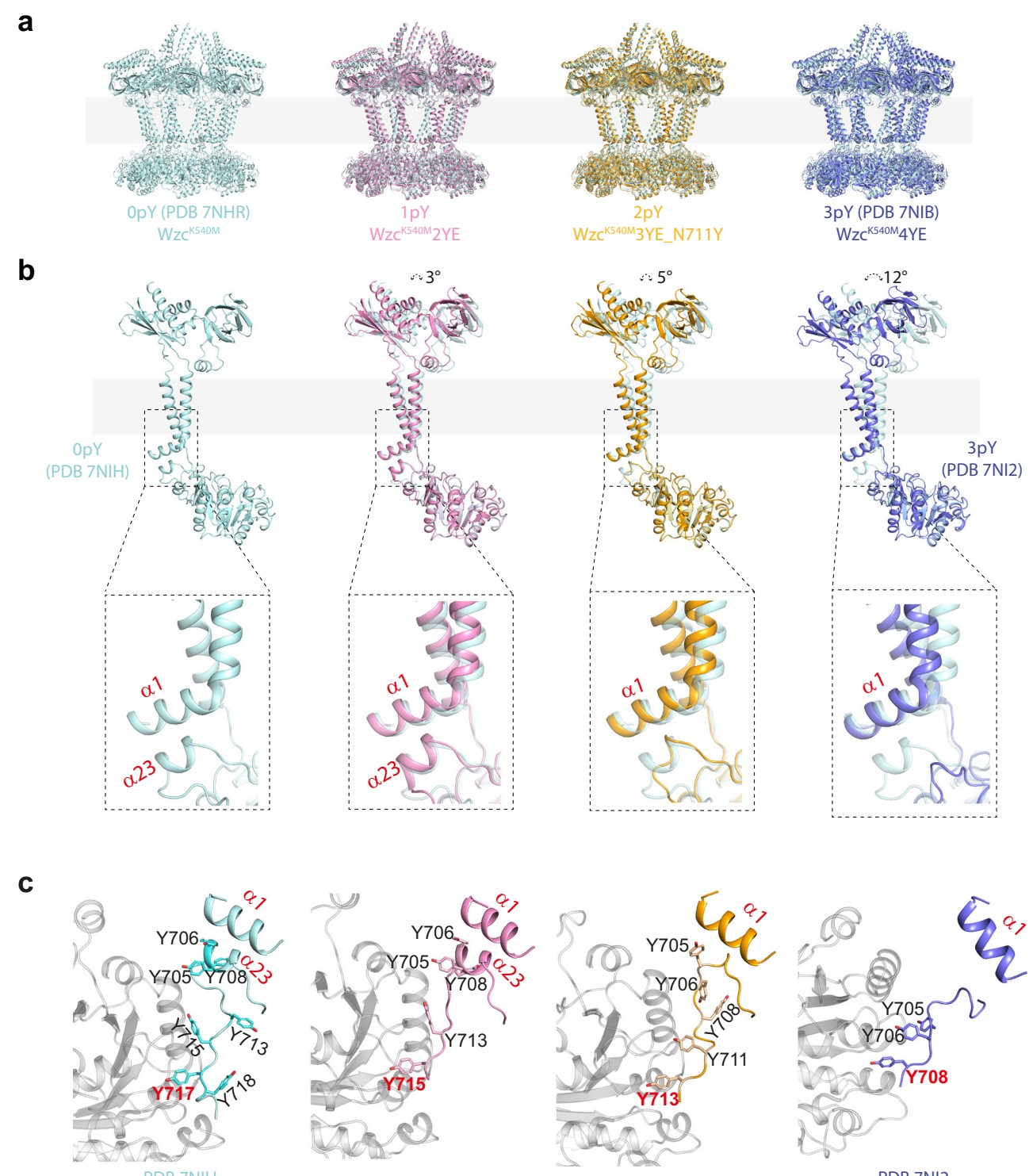

**Fig. 3 | Conformational snapshots of Wzc. a** The structures of the octamers reveal shifts in periplasmic and transmembrane domains relative to Wzc$^{K540M}$. **b** Superposition using the kinase domain of the monomer from each snapshot shows the movement in the transmembrane helices and periplasmic domain. Zoomed insets highlight conformational changes involving helices α1 and α23. **c** Structure of the C-terminal region beginning at residue 700 and helix α1 is highlighted, with the kinase domain of neighbouring monomer coloured grey.

polymerisation[21] (Fig. 1a). The observation that octameric Wzz (a PCP-1 protein) encloses a single Wzy protomer to form a complex[6] implies that octameric Wzc might do the same. Given the activity of Wzy is regulated by PCP proteins (Wzz or Wzc), to promote glycan chain extension, the simplest hypothesis is that the stability of the octamer

and thus the PCP-Wzy complex determines Wzy activity. Here we report experimental structures of mutants of Wzc that correspond to intermediates in the phosphorylation pathway. Together with earlier work[9], these now provide a complete structural description of Wzc octamers, from the most stable (0pY) form to the stability tipping

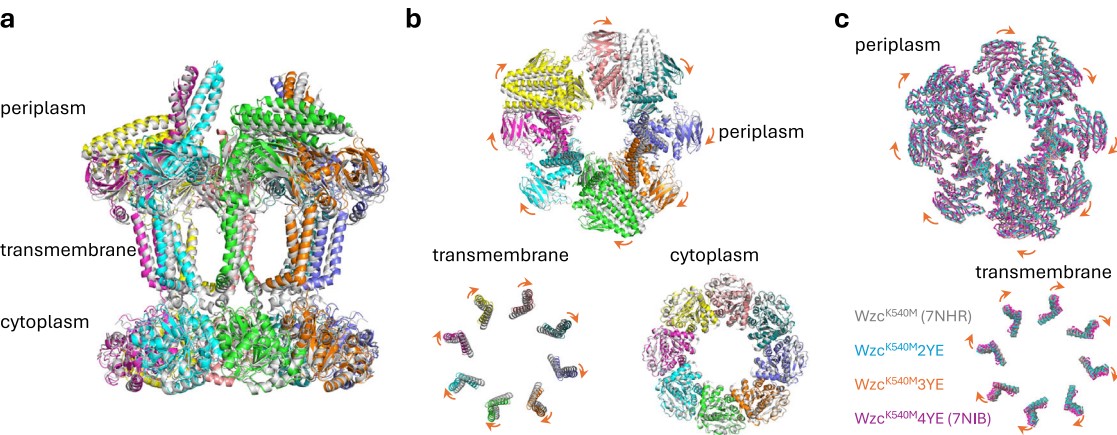

**Fig. 4 | Comparison of the structures of Wzc octamers. a** Side view comparing the structures of Wzc$^{K540M}$ (PDB ID: 7NHR) and Wzc$^{K540M}$4YE (PDB ID: 7NIB). Wzc$^{K540M}$ (PDB ID: 7NHR) is coloured gray, and Wzc$^{K540M}$4YE (PDB ID: 7NIB) is coloured by chain. **c** Structure comparison of the four states. The octameric cytoplasmic kinase domain is used for the structural alignment.

point (3pY). The mutant corresponding to 4pY, "beyond the tipping point", cannot be isolated as an octamer[9]. Mass spectrometry had suggested phosphorylation proceeded sequentially from C to N-terminal in direction[9]. The structural and MD data reported here confirm this hypothesis, showing that the tyrosine most stably positioned at the active site is the one immediately N-terminal to the phosphotyrosine (or the engineered Tyr-to-Glu mutation).

Description of the intermediates provides a molecular description of the structural changes that accompany the progressive phosphorylation of Wzc and reveal the transmembrane signalling mechanism that is essential for CPS production. During the progressive phosphorylation sequence, the kinase domain is a rigid body (Supplementary Fig. 5a). Structural alignment of the kinase domains of monomers of the full-length protein reveals increasing tilting of the transmembrane and periplasmic regions (Fig. 3). When the octamer is viewed from the periplasm to the cytoplasm, progressive phosphorylation results in a clockwise rotation of transmembrane and periplasmic regions relative to the kinase ring (Fig. 4b–c). These rotations are underpinned by changing interactions in the cytoplasmic regions between helix α1 and C-terminal helix α23 (connected to the tyrosine-rich tail). The movement of helix α1 is directly transmitted to transmembrane helix α2 (Figs. 3b, 6c) and then to the periplasmic domain and, in this way, the phosphorylation signal from the cytoplasm is coupled to the periplasm. This general concept of conformational coupling across the membrane is analogous to signaling mechanisms of some transmembrane receptors, such as the histidine kinase of the two-component system in bacteria[22,23] and the EGF receptor in eukaryotes[24,25]. In these proteins, an environmental signal is transmitted into the cell via transmembrane conformational coupling. For Wzc, the information of the cytoplasmic phosphorylation is transmitted to the periplasmic region. In the context of CPS production, the periplasmic region of Wzc is thought to engage with the outer membrane translocon, Wza and genetic[26] and preliminary structural data[27] support Wza-Wzc interactions. Such interactions could conceivably facilitate feedback from Wza to the biosynthesis machinery. *E. coli* mutants lacking Wza exhibit drastic reduction in CPS synthesis[28], consistent with signal across the periplasm that promotes synthesis of CPS only when subsequent translocation of the polymer is possible. Polymerisation does proceed in the absence of translocation in *E. coli* with a mutated Wza that forms stable oligomers but is compromised in its proper insertion into the outer membrane[29]. This is consistent with a hypothesis that it is the formation of Wza-Wzc interface that results in a signal that regulates polymerisation. Gram-positive PCP-2 proteins typically employ two polypeptides that combine to constitute a Wzc equivalent (Supplementary Fig. 11). Analogues to helix α1, helix α23

and ion pair (D28 K701) found in Wzc are conserved in Gram-negative PCP-2, suggesting a common coupling mechanism.

When a phosphotyrosine residue is displaced from the active site the C-terminal peptide becomes mobile. The YxY sequence pattern is highly (but not absolutely) conserved in the BY-kinase family. In Wzc, when Y717 (0pY) is positioned at the active site, Y715 is bound at the tyrosine pocket and is pre-oriented to shift into the active site. When Y715 occupies the active site (1pY), Y713 is bound in the same pocket. When Y708 occupies the active site (3pY, Wzc$^{K540M}$4YE[9]), Y706 also sits at the pocket but with a different relative position. Once Y708 is phosphorylated (4pY), the octamer becomes unstable, consistent with the lack of any phosphorylation of Y706[9]. The presence of the negatively charged E675 in the tyrosine pocket we suggest is incompatible with binding a negatively charged phosphorylated tyrosine (or glutamic acid side chain). One intermediate, 2pY which has Y713 at the active site, does not follow the YxY pattern, since position 711 is asparagine. Notably in the Wzc$^{K540M}$3YE structure the stretch of residues connecting the α23 to the active site is entirely disordered and molecular dynamics of 2pY showed this stretch of residues to be highly mobile (Fig. 5b). We show that it is binding of the i-2 tyrosine to the tyrosine pocket (where i is the number of the residue positioned at the kinase active site) orders the tail between the tyrosine at the active site and helix α23. A mutant of 2pY (Wzc3YE_N711Y) stabilises the C-terminal tail was functional in vivo, restoring immunoreactive CPS to levels approximating the wild type (Fig. 7a). This is consistent with a model in which the more stable the Wzc octamer the higher the Wzy activity. We note the reduction of CPS production by destabilising D28A and K701A mutants further support this model. The Wzc3YE_N711Y shows greater phosphorylation relative to Wzc3YE (Fig. 7b) and proteomic analysis of purified WzcYF_713Y/N711Y reveals both N711Y and Y713 are phosphorylated (Supplementary Fig. 12).

We propose that the YxY pattern serves as a molecular ratchet, that by selecting a tyrosine over phosphotyrosine, ensures phosphorylations in unidirectional (moving from the C to the N). In addition, the pocket prepositions its bound tyrosine to translate to the active site (Fig. 7d). The tyrosine pocket is found in both *E. coli* K-12 (PDB ID: 3LA6) (Supplementary Fig. 5b, e) and CapAB$^{K55M}$ (PDB ID: 2VED) (Supplementary Fig. 5c, f), but, as yet, phosphorylated intermediates for these species have not been described. The conservation of the YxY motif and tyrosine pocket, suggests the ratchet is also a conserved feature of the PCP-2 system.

The CPS assembly mechanism described in *E. coli* K30 is also widespread in *Klebsiella pneumoniae*, with the loci likely being transferred by horizontal gene transfer. In hypervirulent *K. pneumoniae*

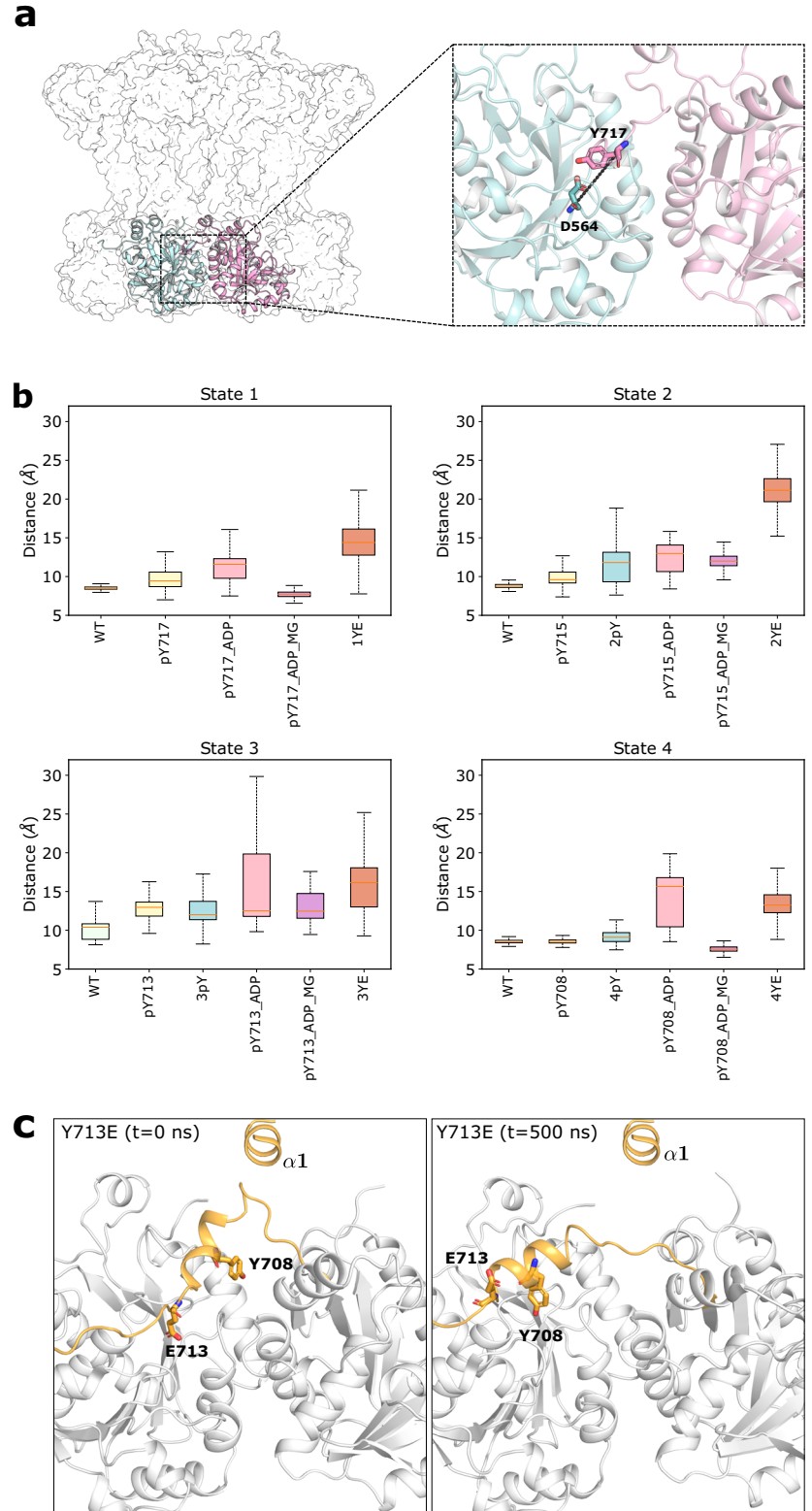

**Fig. 5 | Molecular dynamics analysis of phosphorylation. a** The portion of the structure used in the MD simulation and the metric used in comparison. The metric was defined as the distance between the carbon beta of the tyrosine (or glutamic acid) and the carbon alpha of D564, located in the active site of the adjacent subunit. **b** Boxplots of this metric for each structure. The boxplots represent the variation of the distance over time and the replicates ($n$ = 3000 MD simulations frames. The box bounds the interquartile range divided by the median, with the whiskers extending to a maximum of 1.5 times the interquartile range beyond the box). See Supplementary Table 2 for the summary of the MD simulations, Supplementary Fig. 7 for the time trace. **c** Comparison between the initial and final frames of the 3YE (2pY) showing the displacement of E713 from the active site and the binding of Y708. Helix α1 is shown for reference but was not included in the simulations. Source data are provided as a Source Data file.

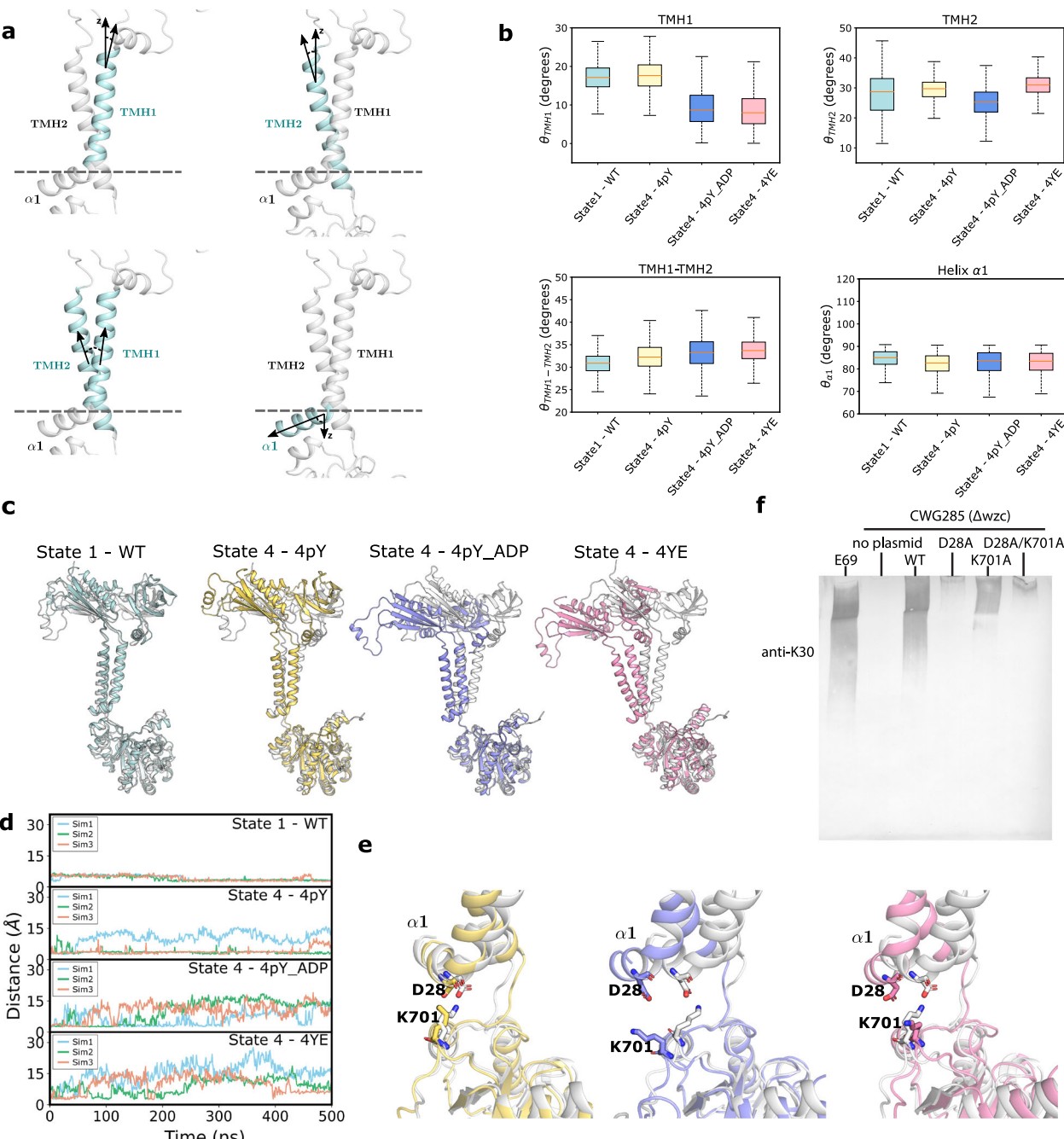

**Fig. 6 | Mechanism of the transmembrane signal transduction. a** The $\theta_{TMH1}$, $\theta_{TMH2}$, $\theta_{\alpha1}$ angles are defined as the angles relative to the axis normal to the membrane, and $\theta_{TMH1-TMH2}$ is the angle between TMH1 and TMH2. **b** Boxplot of the metrics for each system. The boxplots represent the variation of the angles over time and the replicates ($n = 1500$ MD simulations frames. The box bounds the interquartile range divided by the median, with the whiskers extending to a maximum of 1.5 times the interquartile range beyond the box). **c** Superposition (using the kinase domain) of the initial (white) and final frames (coloured) reveals structural shifts for each state of Wzc. **d** Time trace of the minimum distance between the residues D28 and K701. **e** Close up view showing the changes in the interactions between K701 and D28. See Supplementary Table 2 for the summary of the MD simulations. Source data are provided as a Source Data file. **f** Immunoblot (anti-K30 CPS) of whole cell lysates separated by SDS-PAGE. The wildtype *E. coli* strain E69 (serotype O9a:H12:K30) was used as a positive control, and *E. coli* strain CWG285 (Δ*wzc*) was used as a negative control. WT, D28A, K701A, D28A/K701A represent plasmids encoding wild-type Wzc, Wzc with the D28A mutation, Wzc with the K701A mutations, and Wzc with the D28A/K701A mutations respectively. As CPS does not migrate strictly according to size, protein standard is not useful to indicate the molecular weight. These data are representative of three biological replicates.

(HvKP) isolates, the typically heterogenous CPS chain-lengths in non-HvKp isolate are replaced by homogeneous long chain polymers resulting in a hypermucoviscous phenotype evident in colony morphology and high viscous cultures in broth[30]. This property is imparted by an additional protein designated RmpD that interacts with Wzc. While the mechanism underlying RmpD function is not yet known,

based on our data suggesting stability of the Wzc octamer is key to polymerisation activity, we propose RmpD most likely acts to influence stability of the Wzc Wzy complex.

Control of polymerisation is a feature of the chemistry of living things that is often hard to replicate in the laboratory. The structural snapshots here help advance our understanding of how Wzc regulates

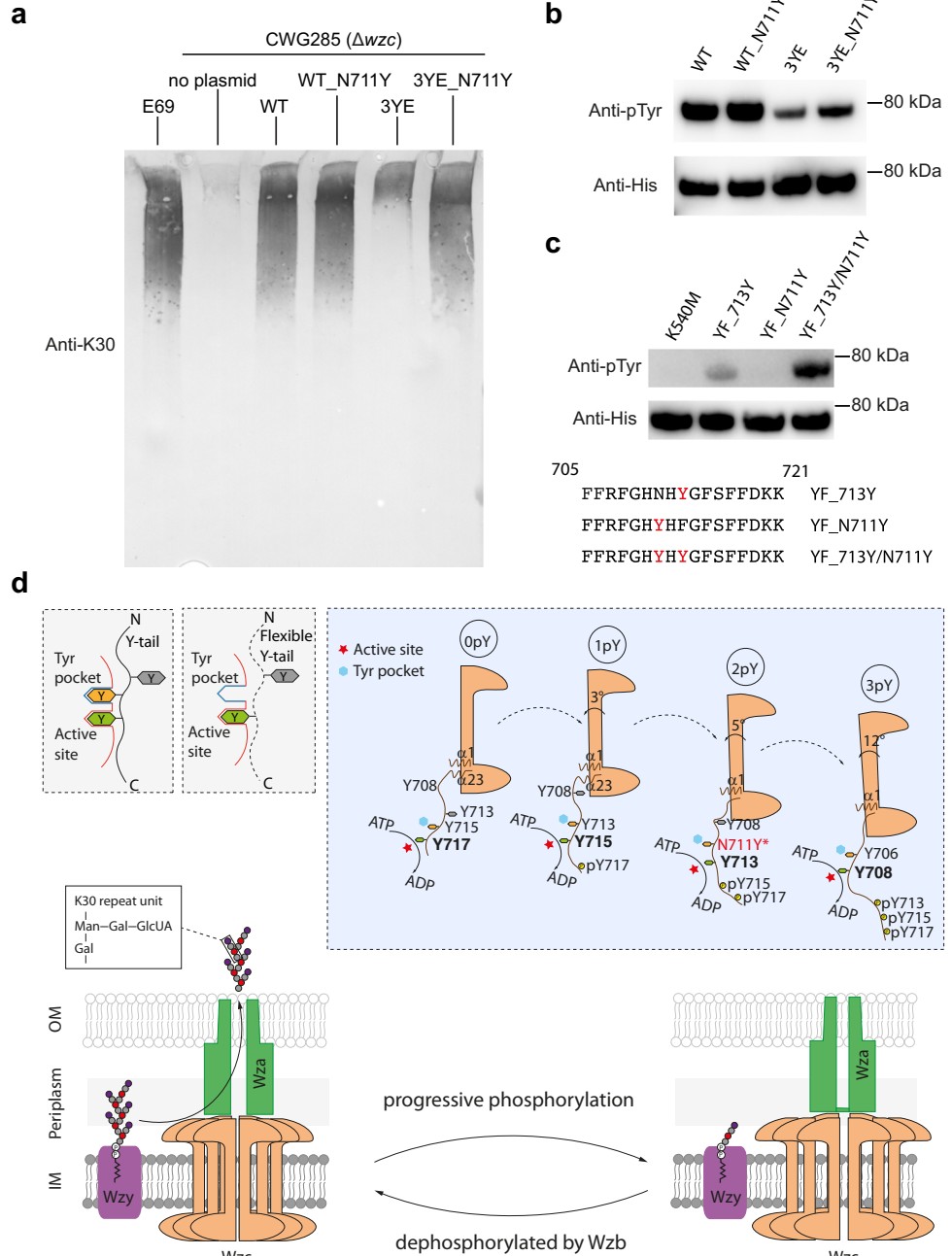

**Fig. 7 | The YxY motif regulates the activity of Wzy. a** Immunoblot (anti-K30 CPS) of whole cell lysates separated by SDS-PAGE. The wildtype *E. coli* serotype K30 strain (E69) was used as a positive control, and *E. coli* CWG285 (Δ*wzc*) was used as a negative control. WT, WT_N711Y, 3YE, 3YE_N711Y represent plasmids encoding wild-type Wzc and derivatives with the N711Y, Y715E/Y717E/Y718E and Y715E/Y717E/Y718E/N711Y mutations, respectively. As CPS does not migrate strictly according to size, protein standard is not useful to indicate the molecular weight. These data are representative of three biological replicates. **b** Corresponding western immunoblot of purified Wzc proteins probed with anti-pTyr antibody (top) and anti-hexahistidine antibody (bottom). The experiment represents one of three technical replicates. **c** Western immunoblot of purified Wzc proteins using anti-pTyr antibody (top) and anti-hexahistidine antibody (bottom). The mutations are identified above each lane. The C-terminal sequences of the tyrosine-rich portion of YF_713Y, YF_N711Y and YF_713Y/N711Y are shown below. The data shows one of three technical replicates. **d** The ratchet mechanism. The YxY motif acts as a clamp stabilizing the octamer but also ordering the C-terminal tail (top). The tyrosine-rich tails in the octamer are phosphorylated and move unidirectionally through the active sites of adjacent monomers (top right). Phosphorylation progressively destabilizes the octamer which, after the fourth phosphorylation, disassembles (top right and bottom). Wzc regulates the polymerization activity of Wzy (purple) and is thought to form a complex with the translocon Wza (green). The cycling of the structural arrangement of the octamer is driven by (auto)phosphorylation and dephosphorylation (catalysed by Wzb) and the cycling is required for the production and export of CPS (bottom).

CPS polymerisation. CPS (and EPS) are important defence mechanisms against the host immune system for pathogenic bacteria. Since the BY-kinase is conserved among bacteria but is distinct from eukaryotic counterparts, the BY-kinases may be promising targets for antibacterial drug development.

## Methods

### Analysis of K30 CPS production in vivo

Wild-type Wzc and the 3YE mutant derivatives[9] were produced from pBAD24[31] derivatives under control of an L-arabinose-inducible promoter. The N711Y mutant derivatives were constructed by site-directed

mutagenesis and using Gibson assembly following the manufacture's protocol (New England Biolabs). *E. coli* CWG285[14] was transformed with the various plasmids encoding Wzc derivatives and the strains were grown at 37 °C in LB broth (with no added L-arabinose) to an O.D.$_{600nm}$ of 0.6–0.8. L-Arabinose serves as the inducer for transcription of the cloned genes from the vector pBAD promoter, but basal levels of transcription are still observed in the absence of inducer. *E. coli* E69 (O9a:K30) and CWG285 were grown as K30 capsule positive and negative controls, respectively. One OD unit of each culture were collected by centrifugation at $12,000 \times g$ for 2 min and the pellets were lysed in 1× SDS-PAGE loading buffer for 10 min in a boiling water bath. Protein was removed by digestion with 0.5 mg/mL proteinase K at 55 °C for 1 h. Samples (10 μL) were separated by SDS-PAGE (Tris/glycine) on 8% acrylamide gels at 150 V for 70 min using the BioRad Mini-protean system. Capsular polysaccharide was transferred to nitro-cellulose (Amersham Protran 0.45 μm NC, Cytiva) at 350 mA for 45 min in 25 mM Tris, 150 mM glycine, 20 % (v/v) methanol, pH 8.3. Immunoblots were blocked in 5% (w/v) skimmed milk in TBST (10 mM Tris-Cl, pH 7.5, 150 mM NaCl, 0.05% (v/v) Tween-20). Capsular polysaccharide was detected with rabbit anti-K30 antiserum[32] as the primary antibody at 1:3000 dilution and goat anti-rabbit alkaline phosphatase (AP) conjugated antibody (Cedar Lane, CLCC43008, 1:3000) as the secondary antibody. AP was detected with nitroblue tetrazolium and 5-bromo-4-chloro-3-indolyl phosphate (Roche).

**Phosphotyrosine western immunoblot of purified Wzc proteins**
The detection of phosphorylated Wzc proteins was performed as described previously[9]. In total, 8 μl purified Wzc proteins at 0.2 mg/ml were loaded and analysed by western immunoblotting. For the detection of phosphotyrosine, the monoclonal anti-pTyr antibody from mouse (Sigma, P4110, 1:5000 dilution) was used as primary antibody and HRP conjugated anti-mouse IgG antibody (Promega, W4021, 1:5000 dilution) as secondary antibody. The loading amount of Wzc proteins was detected by the anti-polyHistidine-peroxidase antibody (Merck, A7058, 1:5000 dilution). The uncropped blots are shown in Supplementary Fig. 14.

**Expression and purification of Wzc proteins**
Proteins were expressed in *E.coli* TOP10 cells and purified essentially as previously described[9]. Cells were grown at 37 °C until the OD600 was around 0.8 before induction with 0.02% arabinose at 20 °C overnight. Cells were harvested by centrifugation and resuspended with lysis buffer (20 mM Na phosphate, pH 7.0, 500 mM NaCl). Constant Systems cell disruptor was used to break the cells, and unbroken cells were removed by centrifugation at $20,000 \times g$ for 1 h at 4 °C. The supernatant from the previous step was collected for further ultracentrifugation at $186,000 \times g$ for 1 h at 4 °C to obtain the cell membranes. The membranes were solubilized with 20 mM Na phosphate, pH 7.0, 500 mM NaCl, 1.0% DDM at 4 °C and undissolved membranes were removed by ultracentrifugation. The supernatant containing Wzc proteins was loaded to ABT nickel resin (Cat. No. 6BCL-NTANi-100). After successive washes with 20 mM Imidazole and 50 mM Imidazole, the protein was eluted with 20 mM Na phosphate, pH 7.0, 500 mM NaCl, 0.003% LMNG and 300 mM Imidazole. Size-exclusion chromatography was performed for further purification using a Superose 6 10/300 increase column (Cytiva) equilibrated with 20 mM HEPES, 150 mM NaCl, 0.001% LMNG, 2 mM tris(2-carboxyethyl)phosphine (TECP), pH 7.3.

**Cryo-EM analysis**
Purified Wzc proteins (Wzc$^{K540M}$2YE, Wzc$^{K540M}$3YE and Wzc$^{K540M}$3YE_N711Y) at ~2 mg/ml were incubated with 1 mM ADP and 10 mM MgCl$_2$ at 4 °C for 1 h. 3.5 μl sample was applied to glow-discharged Quantifoil gold R1.2/1.3 grids (300-mesh) and the grids were blotted for about 3 sec under 100% humidity and 4 °C before vitrification in liquid ethane using Vitrobot.

The datasets were collected on Krios G4 equipped with cold FEG, Falcon 4 direct electron detector and Selectris energy filter. Data was collected using EPU software at a magnification of ×165,000 with the pixel size of 0.737 Å/pixel. Three datasets for the Wzc$^{K540M}$2YE ADP complex, and two datasets for the Wzc$^{K540M}$3YE ADP complex, were collected and combined for data processing respectively. One dataset for Wzc$^{K540M}$3YE_N711Y ADP complex was collected for data processing. Detailed parameters for data collection were listed in the Supplementary Table 1.

The datasets were processed using cryoSPARC software[33] and detailed workflows are described in Supplementary Figs. Briefly, the EM movies were motion-corrected in cryoSPARC and contrast transfer function values were estimated by CTFFIND[34]. Particles were picked through blob picking, or template picking using the templates generated from a small number of auto-picked particles. Multiple rounds of 2D classification were carried out and good 2D classes were selected for further processing. Multiple rounds of heterogeneous refinement were performed in cryoSPARC. Good classes were selected for further non-uniform refinement[35] to get final maps, either without applying symmetry (C1) or with C8 symmetry.

**Model building and refinement**
Coot was used for the manual buildings and adjustments for the PDBs of Wzc$^{K540M}$2YE, Wzc$^{K540M}$3YE and Wzc$^{K540M}$3YE_N711Y. Ligand restraints were generated by eLBOW within PHENIX[36]. Structure refinements were carried out using PHENIX. PyMOL (The PyMOL Molecular Graphics System, Version 2.2.0 Schrödinger, LLC), Chimera[37] and ChimeraX[38,39] were used to prepare the figures of cryo-EM maps and structures.

**MD Simulations**
The cryo-EM structures for the WT Wzc octamer[9] and the Tyr-to-Glu mutants were used to build molecular models for the four states of Wzc. The missing residues, including the missing residues in the C-terminal tail for states 2 to 4 were modelled based on the structures obtained from an AlphaFold2 multimer[40]. To study the effect of the phosphorylation on the C-terminal tail dynamics, we performed MD simulations of a dimer of the kinase domains (Fig. 5a). For each state, a series of MD simulations with different phosphorylation levels, in the presence or absence of ADP and Mg$^{2+}$ and for Tyr-to-Glu mutations were performed. The CHARMM-GUI builder[41] was used to set up the systems for the MD simulations. The kinase domain dimers were solvated with TIP3P waters and 0.15 M NaCl was added to neutralise the charge of the systems. The systems were equilibrated for 1 ns maintaining the structure of the protein restrained. Three repeats of 500 ns of MD simulation were then performed with position restraints removed for each system.

To study the role of helix α1 in signal transduction, MD simulations of two adjacent subunits were performed. For these simulations the entire structure of the protomer, excluding motif 3, was used. Four systems were built: 1) phosphorylation 2) triply phosphorylated with Y718 also phosphorylated; 3) triply phosphorylated with Y718 also phosphorylated bound to ADP and 4) Wzc4YE. The CHARMM-GUI membrane builder[41] was used to set up the systems for the MD simulations. The systems were inserted in a bilayer containing 25% 1-palmitoyl, 2-oleoyl phosphatidylglycerol and 75% 1-palmitoyl, 2-oleoyl phosphatidylethanolamine and solvated with TIP3P waters and 0.15 M NaCl. The input files for minimization and equilibration provided by CHARMM-GUI were used. The system was first minimized followed by a series of NVT and NPT equilibration steps consisting of gradual removal of the restraints from lipid and protein atoms for a total time of 2 ns. Three repeats of 500 ns of unrestrained atomistic MD simulations, for each configuration of the molecular system were performed.

MD simulations were performed using GROMACS 2021[42] and CHARMM36m force field[43] with a timestep of 2 fs. All simulations were

performed at 300 K and 1 bar with protein, lipids and solvent separately coupled to an external bath, using the velocity-rescale thermostat[44] and Parrinello-Rahman[45] for pressure coupling. All bonds were constrained with the LINCS algorithm[46]. The long-range electrostatic interactions were computed with the Particle Mesh Ewald method[47], while a verlet cut-off method was used to compute the non-bonded interactions. MD simulations were analysed using GROMACS tools, MDAnalysis[48] and in-house protocols. Summary of MD simulations were listed in Supplementary Table 2. All images were generated using PyMOL (The PyMOL Molecular Graphics System, Version 2.5.7 Schrödinger, LLC).

## Proteomics analysis

An SDS-PAGE gel band for purified WzcYF_713Y/N711Y was sliced, destained, reduced with 50 mM TCEP for 10 min at 60 °C, alkylated with 100 mM iodoacetamide (IAA) at room temperature for 30 min in the dark, and incubated with trypsin at 37 °C overnight for proteolytic digestion. Digested peptides were dried using a SpeedVac vacuum concentrator (Thermo Fisher Scientific) and resolubilized with 100% $H_2O$ with 1% formic acid for LC-MS/MS analysis. The tryptic peptides were analyzed on a Dionex Ultimate 3000 UHPLC coupled to an Orbitrap Eclipse Tribrid mass spectrometer (Thermo Fisher Scientific). The peptides were firstly loaded onto a 75 μm ×2 cm precolumn and separated on a 75μm × 15 cm Pepmap C18 analytical column (Thermo Fisher Scientific) with a binary buffer system. Buffer A was 0.1% formic acid (FA) in 100% $H_2O$ and buffer B was 0.1% FA in 80% acetonitrile with 20% $H_2O$. Briefly a linear gradient of 0–40% buffer B was used to separate the peptide. The Eclipse mass spectrometer was operated in positive ion mode using data-dependent acquisition with a 3 s cycle time. Precursors and products were detected in the Orbitrap analyzer at a resolving power of 60,000 and 30,000 (@ m/z 200), respectively. Precursor signals with an intensity > $1.0 \times 10^4$ and charge state between 2 and 7 were isolated with the quadrupole using a 0.7 m/z isolation window (0.5 m/z offset) and subjected to MS/MS fragmentation using higher energy collision-induced dissociation (30% relative fragmentation energy). MS/MS scans were collected at an AGC setting of $1.0 \times 10^4$ or a maximum fill time of 100 ms and precursors within 10 ppm were dynamically excluded for 30 s. Raw data files were processed using Proteome Discover 3.0 (Thermo Fisher Scientific) against a sequence database containing sequence of WzcYF_713Y/N711Y (the construct used in this study) as well as a list of common contaminants to identify the phosphorylation sites with the following search parameters: trypsin digestion, fixed modification with carbamidomethyl (C); dynamic modifications with oxidation (M), acetylated protein N terminus, and phosphorylation (Y); up to 2 missed cleavage sites, peptide length from 6 to l44; precursor tolerance (10 ppm); fragment mass tolerance (0.6 Da) The mass spectrometry proteomics data have been deposited to the ProteomeXchange Consortium via the PRIDE[49] partner repository with the dataset identifier PXD061798.

## Reporting summary

Further information on research design is available in the Nature Portfolio Reporting Summary linked to this article.

## Data availability

EM maps and models have been deposited in the EMDB and wwPDB under accession codes EMD-50042 and PDB 9EXO (C1 Wzc$^{K540M}$2YE ADP complex); EMD-50043 and PDB 9EXP (C8 Wzc$^{K540M}$2YE ADP complex); EMD-50044 and PDB 9I2Q (C1 Wzc$^{K540M}$3YE ADP complex); EMD-50045 and PDB 9I2R (C8 Wzc$^{K540M}$3YE ADP complex); EMD-50046 and PDB 9EXQ (C1 Wzc$^{K540M}$3YE_711Y ADP complex); EMD-50047 and PDB 9EXR (C8 Wzc$^{K540M}$3YE_711Y ADP complex). All constructs are available from the authors upon request until their deposition and release by ADDGENE (Wzc_N711Y: #217920; Wzc3YE_N711Y: #217919;

Wzc$^{K540M}$3YE_N711Y: #217918; Wzc$^{K540M}$3YE: #217917; Wzc$^{K540M}$2YE: #173451[9]). The MD simulations data generated in this study are available as supplementary dataset on Zenodo (https://doi.org/10.5281/zenodo.15025687). The mass spectrometry proteomics data have been deposited to the ProteomeXchange Consortium via the PRIDE partner repository with the dataset identifier PXD061798. PDB codes of previously published structures used in this study are 3LA6 and 2VED. Source data are provided as a Source Data file. Source data are provided with this paper.

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

## Acknowledgements

The Rosalind Franklin Institute is funded by UK Research and Innovation through the Engineering and Physical Sciences Research Council (EPSRC). Experimental funding is provided by the Wellcome Trust through the Electrifying Life Science (220526/Z/20/Z to J.H.N.). J.L., Y.Y. are also supported by the Fundamental Research Funds for the Central Universities. C.W. was supported by a Canada Research Chair and research funding from the Canadian Institutes of Health Research (FDN-2016-148364). Mass spectrometry work is supported by Medical Research Council programme grant (MR/V028839/1) awarded to C.V.R. P.J.S.'s lab was funded by Wellcome (208361/Z/17/Z), MRC, BBSRC, EPSRC, NIH, JPIAMR and the Howard Dalton Centre. This project made use of time on ARCHER2 granted via the UK High-End Computing Consortium for Biomolecular Simulation, HECBioSim (http://www.hecbiosim.ac.uk), supported by EPSRC (grant no. EP/R029407/1). P.J.S. would like to thank the SCRTP at Warwick for use of the computing infrastructure.

## Author contributions

Y.Y. purified the proteins. Y.Y., J.L., and J.H.N. carried out an EM analysis. M.B. and P.J.S. carried out MD simulations. B.R.C., M.R.A., N.M.K. and C.W. carried out in vivo analysis. H.S. and C.V.R. carried out mass spectrometry, with the contribution from A.L.B. on sample handling. All authors contributed to the writing of the manuscript.

## Competing interests

The authors declare no competing interests.
