## [Transparent Peer Review file · Nature Communications]

Molecular basis for the phosphorylation of bacterial tyrosine kinase Wzc

Corresponding Author: Dr Jiwei Liu

Version 0:

Reviewer comments:

Reviewer #1

(Remarks to the Author)

In this manuscript, authors Yang et al. solve structures of *E. coli* K30 Wzc in full-length form as Y-to-E mutants to mimic various phosphorylation states of its C-terminal tail. They use these structures and a collection of molecular dynamics (MD) simulations to suggest a mechanism through which the phosphorylation state of the C-terminal tail located in the cytoplasm is communicated through the trans-membrane domains to effect changes in the periplasmic domain. The work is potentially interesting to followers of the field, and to those interested in protein kinases in general. However, I find the analyses and discussion, both regarding the experimental structures and the MD simulations, to be somewhat cursory. I do not feel that the manuscript, in its current form, is suitable for publication in Nature Communications.

Specific comments:

1. Given that the proposed inside-out signaling (deciphering features of the outside-in signaling will require further studies) represents a new paradigm for through-membrane signal transduction coupled to tyrosine phosphorylation, it is perhaps worth adding a few words comparing/contrasting it with the mechanisms active in through-membrane signaling in eukaryotic receptor tyrosine kinases. Perhaps, a good place to start would be in the work of the Kuriyan group on EGFR e.g., Cell 2013, 152, 543; Cell 2009, 137, 1293 etc., specifically with respect to the juxta-membrane regions and the trans-membrane domains. Additionally, given the information they have now, the authors can at least speculate if this mechanism can also be operative in the Gram-positives. I feel that these analyses will enhance the discussion and highlight the uniqueness of what is being described.
2. Given that structures of several phospho-mimetic mutants are available, it is somewhat disappointing that a detailed structural comparison of these various species has not been provided. Are there other differences e.g., within the catalytic elements or elsewhere like the interfaces, beyond the domain rotations and the conformational changes in the alpha1 and alpha23 helices discussed in the text?
3. The tyrosine-specific pocket that appears central to the analysis has not been defined. It is important that the specific features of this pocket are described and illustrated. What are the characteristics of this pocket that makes it selective for a tyrosine over a phospho-tyrosine? Why is a YxY element optimal as suggested?
4. In the model presented the major driver appears to be the interaction between alpha1 and alpha23 centered on the D28-K701 salt-bridge. Could this model be tested through mutations of these residues, perhaps using CPS levels as output? Also, what is the degree of conservation of these residues? This will help define the generality of the mechanism.
5. The authors mention the directionality of phosphorylation from the C- to the N-terminal end of the tail. While it was addressed to some degree in their previous paper, it would be worth mentioning here as well how they arrive at this. It is notable that the sequence of mutations performed here are inherently directional so it may confuse the readers.
6. I am a bit concerned over the species chosen for the MD simulations. Given the arrangement of monomers within the octameric species, a trimeric unit rather than a dimeric unit, where the central monomer likely mimics that in the context of an octameric assembly, is perhaps warranted. Enhanced sampling simulations on the monomeric catalytic subunit have shown extensive coupling of the conformational changes at the active site and the oligomerization interfaces (Science Advances, 2020, 6, eabd3718; Science Advances, 2021, 7, eabj5836). While I am not suggesting that the authors redo all the simulations, 1-2 representative ones to confirm that the inferences for key states are unchanged, would be very useful. Given that the natural state is an octamer, the stability of the dimeric, and proposed trimeric forms during the course of the simulation should also be illustrated e.g., through radius of gyration traces or some other similar global measures.
7. In Figure 4b, it is unclear what the "error bars" on the box plots represent – are these variations over the course of the trajectory or over the replicates? Why not represent these as probability distributions if it is the former for each replicate? That would be much more informative. Also, it is evident that a pTyr is closer to WT and the Y-to-E mutant is not really

representative of a pTyr – this is most evident in states 2 and 4. So it is not fully clear what is being illustrated.

8. As in the case of Figure 4b, Figure 5b is also confusing. It is unclear what the error bars represent in this case as well. Further, perhaps the authors should represent flexibility of the C-terminal tail around the active site (mentioned in line 174, line 176 etc.) using a time-trace of an appropriately defined segment. A comparison of initial and final frames is not a robust representation of flexibility since one or the other could represent a rare state, and dynamics on a fast timescale (sampled extensively in the MD) is anticipated. Also time-traces of the critical salt-bridge should be shown in all cases to convince readers of the disengagement proposed.

9. I feel that an additional control for the data in Figure 6c is needed. It is worth convincing oneself, perhaps through LC-MS/MS that Y713 is the only site phosphorylated in the YF_713Y/N711Y construct given that its properties could be different from the YF_N711Y construct used as control.

Reviewer #2

(Remarks to the Author)

Here, Yang et al. use cryo-em and molecular dynamics to visualize structural snapshots of the octameric Wzc complex involved in regulating CPS production. In this paper by mimicking phosphorylation with Y to E mutations of the tyrosine rich c-terminal tail they capture 2 intermediate conformations. Together with their previous paper (Nat Commun 12, 4349 (2021)) they claim these 2 new structures now complete the set of intermediates of sequential phosphorylations of the c-terminal tyrosines. Based on this the authors propose a ratchet mechanism with a second tyrosine binding pocket adjacent to the kinase active site that helps to stabilize as the c-terminus is pulled through active site sequentially phosphorylating the tyrosines and thus destabilizing the entire octameric complex.

Overall the paper is easy to read and understand. The authors' proposed ratchet mechanism is interesting and seems logical and is supported by their N711Y mutation. Other than that, the paper seems a bit short and vague, and lacks a deeper/better analysis of the data or presentation of the results. The proposed ratchet mechanism alone is nice and warrants publication, however the authors do not convincingly show how this translates to the periplasmic domain other than the 3 and 5° tilts shown in this paper (and 12° shown in their previous paper). The authors do not show the broader consequences to the complex of this tilting, if any. If the authors can connect the ratchet mechanism in the kinase domains to movement in the periplasmic domain with their experimental structures in a convincing way this could really increase the impact of these results.

Some comments and suggestions are listed below:

- It would be nice to have some kind of connection to the periplasmic domain. For example: Do you see any conformational changes in the periplasmic domain in these “snapshots”? From the alignments in fig.3 it looks like there is a rigid-body shift of the entire domain. If all 8 protomers are tilting by 3-12° shouldn't the diameter of the central cavity and/or the entire complex become wider as the degree of tilt gets bigger? Or an opening of the complex (like opening flower pedals maybe?). A 5° tilt multiplied by 8 protomers should add up to quite a big change to the full octamer. Or is something else compensating to maintain the octameric complex? Perhaps some kind of interaction analysis between subunits? If the whole point of the phosphorylation of the tyrosines is to destabilize the complex, then as the tilt increases you should start to lose connections between adjacent protomers. Something like pulling on a loose string causes your sweater to unravel.
- Why no model(s) were built for the 3YE mutant? The stated resolution is quite good... I guess because 3YE_N711Y is virtually identical except for the tail? do you see the same 5° tilt as in 3YE_N711Y? What would an N711Y mutation look like without the 3YE? Or a N711E mutation?
- For the 3YE mutant did you try focused refinement(s) around the kinase domains to improve the density to unambiguously identify the tyrosine in the active site? Or other tools like DeepEMhancer to try and improve the density?
- Similarly, in general there is quite a large number of particles that contribute to your final maps. Have you tried 3D variable analysis or more extensive 3D-classifications to separate or visualize additional subtle conformational changes? Or even to discard missed bad particles to further improve your final maps? Perhaps 3DVA could be used to validate/compliment the MD simulations?
- Could you align your experimental structures with the MD simulations? Do they match?
- If Y718 is not conserved or essential for function why are you mutating it to simulate phosphorylation in your double/triple mutants? Are you not adding additional negative charge where it would not normally be? It is not clear why 2YE = 1pY, 2pY=3YE and 3pY=4YE if in fact Y718 is “phosphorylated”.
- You discuss changes in the interaction between K701 and D28 relating to transmembrane signal transduction. Could the communication between the periplasmic domain and kinase domains be disrupted with mutations to either or both of these residues?
- Second paragraph in the discussion: the last few sentences about Wzz seems out of place, and not really relevant to this paper.
- the first 4 supplementary figures are missing legends.

Version 1:

Reviewer comments:

Reviewer #1

(Remarks to the Author)

The authors have addressed all my concerns satisfactorily. I have no additional comments or concerns.

Reviewer #2

(Remarks to the Author)

In general, the manuscript is much improved compared to the first version. The authors have adequately addressed most of my previous questions.

I have a few comments/questions:

The D28 and K701 mutations seem to support the authors mechanism. But is protein stability and expression levels of these mutants similar compared to WT? If the 2YE, 3YE, and 4YE mutants are, then I guess D28 and K701 mutants should be also? You show the D28-K701 interaction breaking in the MD simulations. But you don't show this interaction in the experimental cryo-em structures. Is this interaction breaking in the experimental structures also? Since this interaction seems to be key to the authors model it would be nice to show it in the experimental structures also. Are the side-chains resolved in this region? in at least the wild-type (0YE)?

Connecting the phosphorylation in the cytoplasmic domain to the full octameric complex to show a twisting of the TM and periplasmic domains is indeed useful. In supplementary figure 11 however you show only the structures from your previous paper. I guess the 2YE and 3YE structures solved here should fit in between those? Also, why is this buried in a supplementary figure? Given that the natural state is an octamer, it might be nice for the reader to see what the effects of phosphorylation are on the full octameric complex more prominently in the article somehow? Or the authors don't feel this aspect is so important?

Technical question: in supplementary table 1 why is there more than 1 value written in the electron exposure row for some of the structures? Were multiple data collections performed and combined? You don't mention anything in the methods section.

We thank the reviewers for the helpful comments. We have attempted to address the reviewers' comments and recommendations with new experimental data, changes to the manuscript text, inclusion of new figures and additional discussion of structural detail. We have now more fully discussed other cross membrane coupling systems. Point by point responses are provided below in red.

Reviewer #1 (Remarks to the Author):

In this manuscript, authors Yang et al. solve structures of E. coli K30 Wzc in full-length form as Y-to-E mutants to mimic various phosphorylation states of its C-terminal tail. They use these structures and a collection of molecular dynamics (MD) simulations to suggest a mechanism through which the phosphorylation state of the C-terminal tail located in the cytoplasm is communicated through the trans-membrane domains to effect changes in the periplasmic domain. The work is potentially interesting to followers of the field, and to those interested in protein kinases in general. However, I find the analyses and discussion, both regarding the experimental structures and the MD simulations, to be somewhat cursory. I do not feel that the manuscript, in its current form, is suitable for publication in Nature Communications.

We have considerably expanded our analysis of experimental structural data and the MD simulations. We did not intend to be cursory and were conscious of word counts, we accept we got the balance wrong in the first submission.

Specific comments:

1. Given that the proposed inside-out signaling (deciphering features of the outside-in signaling will require further studies) represents a new paradigm for through-membrane signal transduction coupled to tyrosine phosphorylation, it is perhaps worth adding a few words comparing/contrasting it with the mechanisms active in through-membrane signaling in eukaryotic receptor tyrosine kinases. Perhaps, a good place to start would be in the work of the Kuriyan group on EGFR e.g., Cell 2013, 152, 543; Cell 2009, 137, 1293 etc., specifically with respect to the juxta-membrane regions and the trans-membrane domains. Additionally, given the information they have now, the authors can at least speculate if this mechanism can also be operative in the Gram-positives. I feel that these analyses will enhance the discussion and highlight the uniqueness of what is being described.

We agree with this helpful suggestion. The discussion has been revised accordingly (Pages 11 - 12, lines 258 - 284).

We now highlight conformational coupling across the membrane is a common thread of typical HKs, EGFR and Wzc. We highlight the distinctiveness of Wzc which

appears to couple Wza in outer membrane to Wzy in the inner bacterial membrane. We discuss the published mutants of Wza that indicate that information from Wza is transmitted to Wzy, via Wzc. The involvement of autokinase and the Wzb phosphatase make the Wzc system particularly complex, perhaps not surprising since the energy consumption of this pathway is high, thus dysregulation will be very deleterious.

The conservation of the signalling mechanism across Gram-negative PCP-2 proteins is fully discussed on Page 12 (lines 283 - 284).

A new Supplementary Figure 12, compares the kinase domains of Gram-positive CapAB^{K55M} (PDB ID: 2VED) and Wzc. We constructed models of Gram-positive PCP-2 proteins using AlphaFold models. The AlphaFold models suggest the basic outlines of the phosphorylation assembly disassembly as a means to regulate Wzy are conserved. However, there are some suggested structural differences. This not surprising given the two-peptide nature of the Wzc system in Gram positive bacteria and the lack of the outer membrane (Wza) partner. It is clear there are further opportunities for study in Gram positive systems.

2. Given that structures of several phospho-mimetic mutants are available, it is somewhat disappointing that a detailed structural comparison of these various species has not been provided. Are there other differences e.g., within the catalytic elements or elsewhere like the interfaces, beyond the domain rotations and the conformational changes in the alpha1 and alpha23 helices discussed in the text?

We agree and have expanded the structural descriptions in the results section (Pages 6 - 8, lines 126 - 178), making clearer the differences and similarities between the different structures. In our revision, we have highlighted the two main differences. It is the critical C-terminal tail which changes most obviously between the structures; this is now outlined in detail. At the global level, these changes are expressed through rotations of the domains relative to each other within the octamer. We show the changes in detail at the C-terminus are connected to domain rotations through the interaction between a1 and a23 helices and the salt bridge (tested by mutagenesis). Within the resolution of the study, we do not identify changes in the positions of the residues at the active site, the tyrosine pocket, or the interfacing residues.

3. The tyrosine-specific pocket that appears central to the analysis has not been defined. It is important that the specific features of this pocket are described and illustrated. What are the characteristics of this pocket that makes it selective for a tyrosine over a phospho-tyrosine? Why is a YxY element optimal as suggested?

We have added the new Supplementary Figure 5 to show the “tyrosine pocket”.

The pocket is described in detail in the results section (Page 7, lines 147 - 149) that describe the new Wzc^{K540M}2YE structure. As noted above, the pocket does not change between the structures.

We are wary of over interpreting the structural data to rationalise the observed preference (N711Y mutant) for tyrosine in the pocket. We do note the aromatic ring of tyrosine could make interactions with the hydrophobic ring of Pro 536 that are not possible for asparagine (Page 8, lines 170 - 172) and presence of the negatively charged E675 in the tyrosine pocket is incompatible with binding a negatively charged phosphorylated tyrosine (or glutamic acid side chain) (Page 13, lines 293 - 295).

The YxY preference arises because of the relative positions of the catalytic site and tyrosine pocket, this is the ratchet.

4. In the model presented the major driver appears to be the interaction between alpha1 and alpha23 centered on the D28-K701 salt-bridge. Could this model be tested through mutations of these residues, perhaps using CPS levels as output? Also, what is the degree of conservation of these residues? This will help define the generality of the mechanism.

This was an excellent suggestion. We constructed mutants of D28-K701 salt-bridge and assessed the functional impact by measuring the CPS production. These new results show that disrupting this contact drastically compromises CPS production, confirming its importance and supporting a model in which the more stable the octamer, the greater the production of CPS. The new experimental results are included into Fig. 5f, and related descriptions are added (Page 10, lines 236 - 238).

In terms of the broader distribution of the mechanism, many Gram-negative Wzc homologs possess negatively-charged residues (D or E) at residue 28 of the helix α 1, and positively-charged residues (K or R) at residue 701 of the helix α 23, comments on the sequence conservation are now made (Page 12, lines 283 - 284).

5. The authors mention the directionality of phosphorylation from the C- to the N-terminal end of the tail. While it was addressed to some degree in their previous paper, it would be worth mentioning here as well how they arrive at this. It is notable that the sequence of mutations performed here are inherently directional so it may confuse the readers.

The referee is right, we did discuss the directionality before based only on mass spectrometry data and this noted in the text.

What is new in this paper is the YxY and tyrosine pocket (experimentally supported by the N711Y mutant which binds at the pocket but N711 does not). This leads to the ratchet mechanism, as the underpinning molecular finding. This paper also has detailed molecular dynamics analysis of the tail in further support of this insight.

6. I am a bit concerned over the species chosen for the MD simulations. Given the arrangement of monomers within the octameric species, a trimeric unit rather than a dimeric unit, where the central monomer likely mimics that in the context of an octameric assembly, is perhaps warranted. Enhanced sampling simulations on the monomeric catalytic subunit have shown extensive coupling of the conformational changes at the active site and the oligomerization interfaces (Science Advances, 2020, 6, eabd3718; Science Advances, 2021, 7, eabj5836). While I am not suggesting that the authors redo all the simulations, 1-2 representative ones to confirm that the inferences for key states are unchanged, would be very useful. Given that the natural state is an octamer, the stability of the dimeric, and proposed trimeric forms during the course of the simulation should also be illustrated e.g., through radius of gyration traces or some other similar global measures.

This is a helpful suggestion to strengthen the manuscript. We have performed additional MD simulations of the trimeric forms of State1 – WT, where the interaction between helices $\alpha 1$ and $\alpha 23$ is strongest, as well as State 4 – 4pY_ADP and State4 – 4YE, where the interaction is absent. These new simulations confirm that the phosphorylation of the Y708 in the presence of ADP increases C-terminal flexibility and leads to disruption of the D28-K701 salt bridge (the new Supplementary Figure 10) (Page 10, lines 229-230).

We also calculated the RMSD of the dimeric and trimeric forms (the new Supplementary Figure 9). Both oligomeric forms exhibit increased flexibility with the addition of phosphorylation/mutation, due the rotation of the transmembrane helices.

The papers cited by the reviewer describe how ATP/ADP and Mg^{2+} affects the conformational states of the catalytic domain of Wzc and its oligomerization. Our simulations probed the effect of phosphorylation/mutation on the stability of the tyrosine within the catalytic binding site. We used the same conformation from the cryo-EM structure in all our analyses but do comment on the bridging role of Mg^{2+} .

7. In Figure 4b, it is unclear what the “error bars” on the box plots represent – are these variations over the course of the trajectory or over the replicates? Why not represent these as probability distributions if it is the former for each replicate? That would be much more informative. Also, it is evident that a pTyr is closer to WT and the Y-to-E mutant is not really representative of a pTyr – this is most evident in states 2 and 4. So it is not fully clear what is being illustrated.

The error bars show the variations over time and over the replicates; boxplots show the distribution of distance for all the conformations sampled in the MD simulations, for each system (structural model probed). Boxplots representing the variation over the time and replicates were chosen for each simulation. Plotting the probability distribution for each replicate would result in 72 different probability distributions. Instead, new Supplementary Figure 7 shows the time trace of the distance between the carbon beta of the tyrosine (or glutamic acid) and the carbon alpha of D564 of the adjacent subunit and its distribution for each simulation.

The pTyr and the Y-to-E mutants both introduce a negative charge. Even though the sizes of the side chains are different, both the Y-to-E mutant and the pTyr promote the same displacement of the residue from the binding site (Supplementary Figure 7). No displacement was observed with tyrosine.

We agree that glutamate is not the same as pTyr, but it is a useful way to gain sight and has been used by others in other systems. The increasing structural instability of the Y to E mutants mirrors the wild type (the tipping point for Y to E is also where phosphorylation stops). Our molecular dynamics show both pTyr and E behave similarly. Therefore we believe it is valid (Page 9, lines 210 - 211).

8. As in the case of Figure 4b, Figure 5b is also confusing. It is unclear what the error bars represent in this case as well. Further, perhaps the authors should represent flexibility of the C-terminal tail around the active site (mentioned in line 174, line 176 etc.) using a time-trace of an appropriately defined segment. A comparison of initial and final frames is not a robust representation of flexibility since one or the other could represent a rare state, and dynamics on a fast timescale (sampled extensively in the MD) is anticipated. Also time-traces of the critical salt-bridge should be shown in all cases to convince readers of the disengagement proposed.

The error bars represent the variability of the angles over time and across the replicates. We didn't use the RMSD of the C-terminal tail to evaluate the effect of the phosphorylation. We don't believe it would be informative in this context because the region changes its fold and varies in length between the structures. To complement figures 5b and 5c we included the MD time-trace of the salt-bridge D28-K701 (Figure 5d).

9. I feel that an additional control for the data in Figure 6c is needed. It is worth convincing oneself, perhaps through LC-MS/MS that Y713 is the only site phosphorylated in the YF_713Y/N711Y construct given that its properties could be different from the YF_N711Y construct used as control.

The Wzc3YE_N711Y shows greater phosphorylation relative to Wzc3YE (Fig. 6b) and proteomic analysis of purified WzcYF_713Y/N711Y reveals both N711Y and Y713 are phosphorylated (Supplementary Fig. 13) (Page 13, lines 306 - 307).

Reviewer #2 (Remarks to the Author):

Here, Yang et al. use cryo-em and molecular dynamics to visualize structural snapshots of the octameric Wzc complex involved in regulating CPS production. In this paper by mimicing phosphorylation with Y to E mutations of the tyrosine rich c-terminal tail they capture 2 intermediate conformations. Together with their previous paper (Nat Commun 12, 4349 (2021)) they claim these 2 new structures now complete the set of intermediates of sequential phosphorylations of the c-terminal tyrosines. Based on this the authors propose a ratchet mechanism with a second tyrosine binding pocket adjacent to the kinase active site that helps to stabilize as the c-terminus is pulled through active site sequentially phosphorylating the tyrosines and thus destabilizing the entire octameric complex.

Overall the paper is easy to read and understand. The authors' proposed ratchet mechanism is interesting and seems logical and is supported by their N711Y mutation. Other than that, the paper seems a bit short and vague, and lacks a deeper/better analysis of the data or presentation of the results. The proposed ratchet mechanism alone is nice and warrants publication, however the authors do not convincingly show how this translates to the periplasmic domain other than the 3 and 5° tilts shown in this paper (and 12° shown in their previous paper). The authors do not show the broader consequences to the complex of this tilting, if any. If the authors can connect the ratchet mechanism in the kinase domains to movement in the periplasmic domain with their experimental structures in a convincing way this could really increase the impact of these results.

Some comments and suggestions are listed below:

- It would be nice to have some kind of connection to the periplasmic domain. For example: Do you see any conformational changes in the periplasmic domain in these "snapshots"? From the alignments in fig.3 it looks like there is a rigid-body shift of the entire domain. If all 8 protomers are tilting by 3-12° shouldn't the diameter of the central cavity and/or the entire complex become wider as the degree of tilt gets bigger? Or an opening of the complex (like opening flower pedals maybe?). A 5° tilt multiplied by 8 protomers should add up to quite a big change to the full octamer. Or is something else compensating to maintain the octameric complex? Perhaps some kind of interaction analysis between subunits? If the whole point of the phosphorylation of the tyrosines is to destabilize the complex, then as the tilt increases you should start to lose connections between adjacent protomers. Something like pulling on a loose string causes your sweater to unravel.

A good point. At monomeric level, when we align the kinase domain, the tilts range from 3-12° across the different phosphorylation states. At the octameric level, when we align the octameric kinase ring, we could see the tilting creates rotation around the central axis of the octameric ring. As a result, the size change of the octamer is small. In essence, a loose string (the tyrosine rich tail) at the cytoplasm is turning the wheel at the transmembrane and the periplasmic regions. To clarify this, we included the new Supplementary Figure 11, and added relevant description in the manuscript text (Pages 11 – 12, lines 258 - 268).

- Why no model(s) were built for the 3YE mutant? The stated resolution is quite good... I guess because 3YE_N711Y is virtually identical except for the tail? do you see the same 5° tilt as in 3YE_N711Y? What would an N711Y mutation look like without the 3YE? Or a N711E mutation?

We built the model for Wzc^{K540M}3YE and deposited the PDBs. We have updated Supplementary Table 1. We did gloss over it in the original submission because the tail (the most mechanistically significant part) was disordered.

The conformation of Wzc^{K540M}3YE is very similar to that of Wzc^{K540M}3YE_N711Y (Supplemental Figure 6). This is now made clear (Page 8, lines 174 - 176).

Generating the construct Wzc^{K540M}N711Y and solving its cryo-EM structures would represent a significant undertaking. We believe it would look just like the native structure which determined, that is with Y717 at the active site and Y715 at the pocket. It is possible that the N711Y change would introduce some local distortion. However, this would not disclose any functional insight on the mechanism.

- For the 3YE mutant did you try focused refinement(s) around the kinase domains to improve the density to unambiguously identify the tyrosine in the active site? Or other tools like DeepEMhancer to try and improve the density?

The density was clear there was a tyrosine at the active site, we suspected Y713 was the residue at the active site, all we had to do was identify the residue N-terminal to it. However, it was experimentally ambiguous, and we did not gain sufficient improvement from focused refinement to change this. We felt tools such as DeepEMhancer (which we have used in other studies) for such a small region ran the risk of convincing ourselves that Y713 was at the active site. In any event, the experimental conclusion was unchanged, the tail of the Wzc-K540M-3YE construct is likely highly flexible. This then prompted to think of the YxY motif which interacted with a ratchet (tyrosine pocket) to stabilise and orient the C-terminal tail.

We introduced the additional mutation N711Y to create the YxY pattern and test our hypothesis. The density was unambiguous and readily interpretable with N711Y in the tyrosine pocket supporting the ratchet mechanism.

- Similarly, in general there is quite a large number of particles that contribute to your final maps. Have you tried 3D variable analysis or more extensive 3D-classifications to separate or visualize additional subtle conformational changes? Or even to discard missed bad particles to further improve your final maps? Perhaps 3DVA could be used to validate/compliment the MD simulations?

These are useful suggestions. We have tried 3D variable analysis and extensive 3D-classifications but have not seen significant improvements of the map or useful separations of sub-conformations.

- Could you align your experimental structures with the MD simulations? Do they match?

This is a good question that we comment in more detail below. In short yes but there are differences.

Figure R1. Superposition of the initial (white) and final frames (pink) of the MD simulation of State4-4YE and the cryo-EM structures PDB ID 7NHR (blue) and PDB ID 7NIB (yellow).

The simulation of the full length shows upon phosphorylation conformational changes are triggered for the TM and the periplasmic regions. The simulations show similar trends in conformational changes to experimental structures with 4YE having the largest movement (Fig. 5e).

The experimental structures Wzc^{K540M} (7NHR) and $Wzc^{K540M}4YE$ (7NIB) represent the unphosphorylated form and the tipping point of the octamer (respectively). We

carried out the alignment of these two experimental structures and with initial and final frames of the simulation of Wzc4YE (Figure R1). The initial frame adopted the conformation similar to Wzc^{K540M} and the final frame shows a large conformational change similar to (but larger than) Wzc^{K540M}4YE (Figure R1). (The larger movement we think reflects the smaller model necessary for efficient computation).

- If Y718 is not conserved or essential for function why are you mutating it to simulate phosphorylation in your double/triple mutants? Are you not adding additional negative charge where it would not normally be? It is not clear why 2YE = 1pY, 2pY=3YE and 3pY=4YE if in fact Y718 is “phosphorylated”.

This is a very valid question.

In the structure of Wzc^{K540M} (PDB ID: 7NHR), there are no Y to E mutations, the residue inserted into the active site is Y717 NOT Y718. However, Y718 can be phosphorylated in a portion of the native protein, noted and discussed in our previous paper’s mass-spectrometry analysis. The residue is not conserved in other Wzc homologues. It was originally mutated to eliminate the risk of any heterogeneity in our studies. Since this worked well, its mutation had no functional or structural consequence (shown in previous paper) and the residue was ordered, we did not feel it was experimentally worthwhile to redetermine all the phosphorylation mimic structures keeping Y718. The nomenclature 2YE = 1pY arises from the fact that the Y718 phosphorylation is not mechanistically required.

- You discuss changes in the interaction between K701 and D28 relating to transmembrane signal transduction. Could the communication between the periplasmic domain and kinase domains be disrupted with mutations to either or both of these residues?

An excellent point and one made the other reviewer. We have made and tested these mutants (see above point 4 in response).

- Second paragraph in the discussion: the last few sentences about Wzz seems out of place, and not really relevant to this paper.

We moved it to the discussion on Page 11 (lines 242 – 246) and more clearly explain relevance.

- the first 4 supplementary figures are missing legends.

Added.

We thank the reviewers for the helpful comments. Point by point responses are provided below in red.

Reviewer #1 (Remarks to the Author):

The authors have addressed all my concerns satisfactorily. I have no additional comments or concerns.

Thank you.

Reviewer #2 (Remarks to the Author):

In general, the manuscript is much improved compared to the first version. The authors have adequately addressed most of my previous questions.

I have a few comments/questions:

The D28 and K701 mutations seem to support the authors mechanism. But is protein stability and expression levels of these mutants similar compared to WT? If the 2YE, 3YE, and 4YE mutants are, then I guess D28 and K701 mutants should be also? You show the D28-K701 interaction breaking in the MD simulations. But you don't show this interaction in the experimental cryo-em structures. Is this interaction breaking in the experimental structures also? Since this interaction seems to be key to the authors model it would be nice to show it in the experimental structures also. Are the side-chains resolved in this region? in at least the wild-type (0YE)?

The complementation experiments were performed using only a very small amount of protein expression to better mimic the natural situation. This amount of protein is below the limit of detection in western blots of cell lysates in our mutant strain background (This is shown in Figure R2-1a).

In order to show mutant proteins could be produced and stability of the variants was unchanged, we performed western blots of whole-cell lysates from *E. coli* TOP10 (for overexpression) transformed with plasmids encoding Wzc and its variants with C-terminal hexahistidine tag. No detectable differences were observed in protein expression level and stability among the WT, D28A, K701A and D28A/K701A (Figure R2-1b).

In the cryo-EM structures of the 0YE - 4YE octamers, the electron densities of the D28 K701 side-chains are resolved, within the resolution of the structures, the distance between D28 and K701 does not change. The simulation shows the breaking of the D28-K701 contact as it goes beyond 4YE that leads to the disassembly of the octamer.

Figure R2-1. (a) Western immunoblot of whole cell lysates from *E. coli* CWG285 (Δwzc) containing plasmids encoding Wzc and its variants with C-terminal hexahistidine tag as indicated above the lanes. The cells were prepared in the same way as those used in Fig. 6f. Protein expression was detected using the mouse anti-pentaHis antibody as the primary antibody (1:3000 dilution) and the alkaline phosphatase conjugated goat anti-mouse antibody (1:3000 dilution) as the secondary antibody. **(b)** For overexpression, the plasmids encoding Wzc and its variants with C-terminal hexahistidine tag were transformed into *E. coli* TOP10 cells and whole cell lysates with and without induction of arabinose were analysed. Protein expression was detected using mouse anti-His antibody (1:10000 dilution) as the primary antibody and HRP conjugated goat anti-mouse IgG antibody (1:10000 dilution) as secondary antibody.

Connecting the phosphorylation in the cytoplasmic domain to the full octameric complex to show a twisting of the TM and periplasmic domains is indeed useful. In supplementary figure 11 however you show only the structures from your previous paper. I guess the 2YE and 3YE structures solved here should fit in between those? Also, why is this buried in a supplementary figure? Given that the natural state is an octamer, it might be nice for the reader to see what the effects of phosphorylation are on the full octameric complex more prominently in the article somehow? Or the authors don't feel this aspect is so important?

We moved these data into the new Figure 4, and added Fig. 4c to show the four states. We have retained panels (a) and (b) showing only the two extremes, because

these emphasize the twisting for the reader. Related descriptions are added to the manuscript (Page 9, lines 194 – 196).

Technical question: in supplementary table 1 why is there more than 1 value written in the electron exposure row for some of the structures? Were multiple data collections performed and combined? You don't mention anything in the methods section.

Yes multiple data collections were collected, this is now stated in the Methods section (Page 17, lines 392 - 395).